# Assessment of Heavy Metals and Trace Elements in the Human Milk of Women Living in Latvia and an Evaluation of Influencing Factors [note 1]

**DOI:** 10.3390/nu16111568

**Published:** 2024-05-22

**Authors:** Līva Aumeistere, Alīna Beluško, Inga Ciproviča

**Affiliations:** Faculty of Agriculture and Food Technology, Latvia University of Life Sciences and Technologies, Lielā iela 2, LV-3001 Jelgava, Latviainga.ciprovica@lbtu.lv (I.C.)

**Keywords:** heavy metals, trace elements, ICP-MS, pollution, lactation period, diet

## Abstract

During lactation, heavy metals and trace elements can be mobilised from the maternal body stores and excreted via human milk. A total of 66 mature human milk samples were collected from lactating women in Latvia between 2016 and 2017 to analyse the content of As, Cd, Pb, Al, Sn, and Ni. Additionally, 50 mature human milk samples were collected between 2022 and 2023 to analyse the content of Cd and Pb. The content of heavy metals and trace elements in human milk was determined using ICP-MS. Only two individual human milk samples contained heavy metals above the method’s detection limit—one with an arsenic content of 0.009 mg kg^−1^ and one with a lead content of 0.047 mg kg^−1^. The preliminary data show that human milk among lactating women in Latvia contains only insignificant amounts of heavy metals and trace elements. Concern over such content should not be a reason to choose formula feeding over breastfeeding. Nevertheless, heavy metals, trace elements and other pollutants in human milk should be continuously monitored.

## 1. Introduction

Human milk is the preferred nutrient source for infants in the first six months of life [1]. During lactation, both nutrients and pollutants can be mobilised from the maternal body stores and excreted via human milk [2]. Infants are especially susceptible to pollutants because their organs, especially the nervous system, are still developing [2].

Heavy metals (arsenic, cadmium, and lead) and trace elements (tin, aluminium, and nickel) are dispelled all over the environment and can be ingested via food and water [3,4,5,6,7,8].

Food and drinking water are the primary sources of exposure to arsenic, which is classified as carcinogenic to humans [3]. Fortunately, only small amounts of arsenic pass through mammary glands to human milk. The median arsenic concentration in human milk is reported to be around 0.31 μg L^−1^, but the maximum is around 0.62 μg L^−1^ [3,9].

The primary sources of cadmium include smoking and food [4]. Cadmium is primarily toxic to the kidneys but can also cause bone demineralisation. Therefore, the European Food Safety Authority advises that the tolerable weekly intake should be no more than 2.5 μg kg^−1^ of body weight per day to ensure protection for all consumers, including vulnerable subgroups like infants [4]. The average reported cadmium concentration in human milk is around 2 μg L^−1^ [10].

Lead is a contaminant that occurs in the environment naturally and from anthropogenic activities such as mining, smelting, and battery manufacturing [6]. Although control measures have been taken to regulate lead in paint, petrol, food cans, and pipes in Europe since the 1970s, exposure to lead still occurs. The primary sources are food, water, air, soil, and dust [6]. Lead, previously accumulated in maternal bones, can be mobilised along with calcium in human milk during lactation [11]. On average, lead exposure in infants via human milk is estimated to be 0.21 μg kg^−1^ of body weight per day [6]. The European Food Safety Authority has set the toxicological reference value BMDL_01_ (benchmark dose limit) for lead at 0.50 μg kg^−1^ of body weight per day, meaning that, in the long term, lead intake ≥0.50 μg kg^−1^ of body weight per day increases the risk of nervous system damage in an infant by one percent compared to the risk level of the unexposed population [6].

The widespread distribution in soil results in the presence of tin within certain foodstuffs, mainly vegetables, fruits, and berries. Because of limited absorption, tin has low systemic toxicity. Acute intake of tin is associated with gastrointestinal symptoms (vomiting, diarrhoea, etc.) [5]. The mean tin concentration in human milk among lactating women from Europe is reported to be lower than 0.50 μg L^−1^ [12,13].

Aluminium occurs naturally in the environment and comes from production processes (papermaking, fire retardants, colours, cosmetics, pharmaceuticals, etc.). Aluminium, mainly in alloys with other metals, is also used for food packaging and cookware [7]. After absorption, aluminium accumulates in various organs and tissues (mostly in bones). Data regarding aluminium toxicity in humans is limited but indicates possible carcinogenic potential, neurotoxicity, etc. [7]. Potential exposure to aluminium in breastfed infants is estimated to be less than 70 μg kg^−1^ of body weight per week [7]. Potential dietary exposure of aluminium from infant formulae and other foods manufactured for infants is reported to be higher—from 100 μg kg^−1^ of body weight per week (for 0−3-month-old infants) to 200 μg kg^−1^ of body weight per week (for 4−6-month-old infants) [7].

Nickel is genotoxic and impairs the function of the immune system [8]. Sources of nickel include emissions from motor vehicles and electric power utilities, as well as food and drinking water. Available data indicate that nickel content in human milk is around 55 μg L^−1^ and, overall, chronic dietary exposure to nickel in infants ranges from around 28 to 30 μg kg^−1^ of body weight per day [8,14].

During the Soviet occupation of Latvia, the industrial sector accounted for 37.1% of the gross domestic product [15]. After the collapse of the Soviet Union in the 1990s, industrial production declined, and electrical and pharmaceutical industries developed. Also, coal and petroleum fuels were replaced with natural gas [15]. All this has led to a decline in environmental pollution in Latvia [15,16]. Since the 1990s, total lead emissions in Latvia have declined by 98.8%, and cadmium has declined by 42.9% [17].

Although the level of pollution in Latvia has significantly decreased in recent decades [15,16,17], it is necessary to continue monitoring pollutant content in Latvia via environmental, food, and human biological samples. Human milk is one of the biological matrices used to evaluate the pollution level. However, the heavy metal and trace element content in human milk among lactating women in Latvia has not been previously analysed. Therefore, this study aimed to evaluate arsenic (As), cadmium (Cd), tin (Sn), lead (Pb), aluminium (Al), and nickel (Ni) content in mature human milk collected from women residing in Latvia. The sample collection was organised into separate periods—from 2016 to 2017 and 2022 to 2023. Additional information was also collected to analyse what influences the content of heavy metals and trace elements in human milk, including the data on dietary habits. 

## 2. Materials and Methods

### 2.1. Selection of Participants and Collection of Human Milk Samples

Before the study, approval by the Riga Stradiņš University Ethics Committee (protocol code 4/28.7.2016. and date of approval—28 July 2016) was received. Ethical approval from the Riga Stradiņš University Ethics Committee was re-established on 13 January 2021 (protocol code 22-2/521/2021). Written informed consent was obtained from all participants. The inclusion criteria for participants were:Reside in Latvia;Singleton pregnancy;At least 28 days postpartum;Exclusively or partially breastfeeding;Mother and child are both healthy (without metabolic disorders, no acute illnesses, etc.).

The exclusion criteria were:Unsigned written informed consent;Non-compliance with the inclusion criteria.

During the study, participants had to collect a sample (~20 mL) of mature human milk (at least 28 days postpartum). It was a pooled sample, and milk was collected from the feeding breast during various feeding sessions throughout the 24 h. Participants could choose the most convenient method for milk expression (by hand, using a breast pump, or combining both methods). The sample was collected in the labelled, graduated 50 mL propylene container (Plastiques Gosselin, Borre, France) and stored in the refrigerator (~4 °C) during the collection process and after was placed in the household freezer (approximately −18 °C). Then, the sample was collected from the participant and transported to the laboratory using a bag with ice packs. Before the analysis, samples were kept frozen (−18 ± 3 °C). In total, 66 mature human milk samples were obtained during the period November 2016−December 2017 to determine As, Cd, Sn, Pb, Al, and Ni content, and, in total, 50 mature human milk samples were obtained during the period January 2022−January 2023 to determine Cd and Pb content. Samples were collected across Latvia without distinguishing a specific region (Figure 1).

Participants also had to complete a food frequency questionnaire about a specific food product and drink intake one month before the study. The food frequency questionnaire consisted of 64 food products and drinks for which the frequency of consumption was estimated by the participants (participants had to mark how frequently specific products had been consumed one month before the study). The frequency of consumption was categorised as follows: Never;Less than once a week;Once a week;Two times a week;Every other day;Every day.

Food products and drinks from the food frequency questionnaire were divided into 18 categories. The categories were:Cereals, cereal products;Bread;Potatoes;Eggs;Meat and processed meat;Milk and milk products;Fish and seafood;Vegetables;Pulses;Fruits and berries;Plant-based fats;Condiments;Sweets and bakery goods;Salty snacks and fast food;Lemonades;Caffeine-containing drinks;Herbal teas;Alcohol.

Information about maternal age, parity, child’s age, sex, birth weight and length, breastfeeding pattern (exclusive or partial), and milk expression manner (by hand, using a breast pump, or combining both methods) was also collected using a questionnaire. 

Overall, 79 participants entered the study from 2016 to 2017, but 13 women dropped out (drop-out rate 16%). In total, 61 participants entered the study from 2022 to 2023, but 11 women dropped out (drop-out rate 18%). Reasons for drop-out were as follows:Non-response to the researcher after agreeing to participate in the study (*n* = 8 in the first study period, *n* = 7 in the second study period);Not being able to collect the necessary amount of human milk for the analysis of the elements (*n* = 1 in the first study period, *n* = 4 in the second study period);The child or the mother developed health problems, and thus they no longer met the inclusion criteria of the study (*n* = 4 in the first study period).

In the end, the study was completed (i.e., human milk samples and questionnaires were collected) by 66 participants between 2016 and 2017 and by 50 participants between 2022 and 2023.

### 2.2. ICP-MS Analysis

Human milk samples for this study were analysed as routine samples in the laboratory providing food testing. The content of heavy metals (As, Cd, Pb) and trace elements (Al, Sn, Ni) was determined using inductively coupled plasma mass spectrometry (ICP-MS) Agilent 7700x (Agilent Technologies, Tokyo, Japan) according to an in-house method BIOR-T-012-148-2013 (developed based on standards EN 15763:2010 [18], ISO 17294-1:2004 [19] and ISO 17294-2:2016 [20]). Collected samples were thawed in hot water (~55 °C) and homogenised using Vortex (VXMNDG Vortex Mini digital, OHAUS Corporation, Parsippany, NJ, USA). Using analytical scales (a weighting capacity of 0.0001 g), 0.3 g of the sample was weighed in the digestion vessel. Then, 2 mL of ultra-pure water, 5 mL of concentrated nitric acid, and 3 mL of concentrated hydrogen peroxide were added using an automatic pipette, and the mixture was held for 10 min. 

Afterwards, the digestion vessel was closed and heated in the microwave oven. The heating time was 15 min with a temperature of 150 °C, then held at that temperature for 15 min. Then, the mixture was heated for 10 min up to 180 °C and held at that temperature for 20 min. After, the vessel was cooled to ≤50 °C.

The vessel was carefully opened, discharging the accumulated gases through a hole in the lid. The sample was filtered through a high-speed filter paper, washing it from the vessel wall with ultra-pure water. The filtered sample was transferred to a 50 mL volumetric flask and filled up to the mark using ultra-pure water.

Measurements were performed according to the ICP-MS Agilent 7700x (Agilent Technologies, Tokyo, Japan) manufacturer’s instructions:Plasma mode—normal, robust;RF forward power (W)—1300;Sampling depth (mm)—8.0;Carrier gas flow (L min^−1^)—0.6;Dilution gas flow (L min^−1^)—0.4;Spray chamber temperature (°C)—2;Extraction lens 1 (V)—0;Kinetic energy discrimination (V)—3.

The test solution, obtained by pressure digestion, was nebulised, and the aerosol was transferred to a high-frequency inductively coupled argon plasma. The high temperature of the plasma was used to dry the aerosol and to atomise and ionise the elements. The ions were extracted from the plasma by a set of sampler and skimmer cones. They were transferred to a mass spectrometer, where the ions were separated by their mass/charge ratio and determined by a pulse-count detector.

Heavy metals and trace elements in all human milk samples were analysed in duplicate. Appropriate element ICP-MS Standards (100 mg L^−1^) (Honeywell Fluka, Charlotte, NC, USA) were used for calibration of ICP-MS Agilent 7700x (Agilent Technologies, Tokyo, Japan). The detection limit of each element was calculated based on the standard deviation of 20 blank samples analysed in triplicate (2.5 mL analytical portion and 50 mL analytical solution). The results from ICP-MS were expressed as mg kg^−1^.

### 2.3. Statistical Analysis

All data were compiled using Microsoft 365 Excel (Microsoft Corporation, Redmond, WA, USA). The results were expressed as mean, standard deviation, and range (minimal–maximal values). To compare the data on heavy metal and trace element concentration in human milk with results from other studies, a unit conversion from mg kg^−1^ to μg L^−1^ was provided (human milk density 1.03 g mL^−1^ [21]).

IBM SPSS Statistics 23 (IBM, Armonk, NY, USA) was used for data statistical analysis. Due to the small sample size, non-parametric tests were used [22]. A Kruskal–Wallis H test was used to evaluate differences between ordinal dependent and independent nominal variables. Spearman’s rank correlation (ρ) was used to measure the strength and direction of the associations between ordinal and continuous variables. A *p*-value of ≤0.05 was considered statistically significant.

## 3. Results

### 3.1. Characteristics of the Participants

All participants completed a questionnaire, providing data about the maternal and child characteristics (Table 1). 

The content of heavy metals and trace elements in human milk in almost all samples was below the detection limit (Table 2). Only two individual human milk samples contained heavy metals above the method’s detection limit—one with an arsenic content of 0.009 mg kg^−1^ and one with a lead content of 0.047 mg kg^−1^, respectively (Table 2).

### 3.2. Dietary Habits among Participants 

All participants completed a food frequency questionnaire, providing data about the intake frequency of specific food products and drinks consumed one month before the study (Figure 2). 

A Kruskal–Wallis H test revealed some significant differences in the dietary habits of participants compared by study periods (Table 3). More frequent consumption of cereals, cereal products, eggs, sweets, and bakery goods but less frequent intake of alcohol was noted among the participants from the study period of 2022–2023.

Breakfast cereal, white wheat pasta, white rice, oat flakes, and buckwheat were the most commonly consumed food products from the cereals and cereal products group—consumed on average once a week. Rye bread, sourdough bread, and seed bread were mostly consumed twice a week, but the average intake frequency for wheat bread was “once a week”. Poultry and pork, on average, were consumed once a week, but beef and meat products (sausages, ham, etc.) were mostly consumed less than once a week. The participants reported that they consumed fish only once a week. Milk, cheese, and sour cream were the most commonly consumed milk products, on average being consumed twice a week. Participants, on average, noted fresh vegetable consumption every other day, but fresh fruit and berry intake was two times a week. Pulses were consumed only rarely. Vegetable oils were the most commonly used plant-based fats, being used every other day. Butter was the most commonly consumed condiment, but chocolate was the most frequently consumed product from the sweets and bakery goods category; both products, on average, were consumed twice a week by the participants. Different salty snacks (chips, salty nuts) and fast food (pizza, hot dogs, etc.) were consumed only rarely, at less than once a week. Lemonades and alcohol were not consumed at all or only rarely. Coffee was the most commonly consumed caffeine-containing drink, being consumed daily by 52% of study participants.

A more frequent fruit and berry intake was observed among participants who had boys (3.19 ± 1.00) compared to participants who had girls (2.81 ± 0.94) (*p* = 0.021). More frequent use of condiments (sauces, mayonnaise, ketchup, jam) was noted among those participants whose children had already started complementary feeding (1.33 ± 0.45), compared to participants who were still exclusively breastfeeding (1.24 ± 0.55) (*p* = 0.033). Similarly, more frequent alcohol consumption was noted among those participants whose children had already started complementary feeding (0.64 ± 0.78) compared to participants who were still exclusively breastfeeding (0.16 ± 0.45) (*p*-value = 0.000). Bread (ρ = −0.187, *p* = 0.049), vegetable (ρ = 0.224, *p* = 0.017), fruit and berry (ρ = 0.203, *p* = 0.032), and plant-based fat (ρ = 0.372, *p* = 0.000) consumption frequency correlated with maternal age, but meat and processed meat (ρ = 0.241, *p* = 0.010) and milk and milk products (ρ = −0.258, *p* = 0.006) intake frequency correlated with child’s age. Maternal sweets and bakery goods intake (ρ = 0.274, *p* = 0.003) and alcohol intake (ρ = 0.214, *p* = 0.024) frequency correlated with the child’s birth length. Parity negatively correlated with salty snacks and fast food (ρ = −0.231, *p* = 0.014) and caffeine-containing drink (ρ = −0.206, *p* = 0.029) intake frequency.

## 4. Discussion

ICP-MS allows us to ascertain elements even in trace amounts. However, the human milk samples for this study were analysed as routine samples in a laboratory providing food testing, with no modifications being used in the method to decrease the detection limits. This makes it difficult to compare obtained values with data reported from other countries (Table 4). Based on the samples where heavy metal concentration was above the detection limit, this study reports higher arsenic and lead values compared to data from Sweden and Spain (Table 4).

Although the obtained results provide reassurance that the content of heavy metals and trace elements in the human milk of women living in Latvia is low, some modifications in the ICP-MS method should be considered to evaluate the pollutant presence more successfully. Acid digestion was used in this study, but Levi et al. (2018) [23] have explored that the precision of ICP-MS analysis could be higher through using alkali dilution compared to acid digestion [23]. The purity of analytical reagents should also be considered [23].

Only two individual human milk samples from this study contained heavy metals above the method’s detection limit—one sample with an arsenic content of 0.009 mg kg^−1^ and one with a lead content of 0.047 mg kg^−1^, respectively. Further investigation was provided, and it was found that the participant, who had the highest lead content in human milk, had previously worked in typography and had left this job because of health issues, possibly related to lead poisoning. Lead can be present in printing inks; therefore, it is important for printing industry workers to use protective equipment (masks, gloves, etc.) to reduce exposure to lead [24]. 

No possible explanation was found for the highest obtained arsenic value. Nevertheless, it should be noted that it was still lower than the maximum levels for arsenic set for the infant formulae and follow-on formulae defined in the Commission Regulation (CE) No. 2023/915 [25] (Table 5). 

Although the health of the mother and child has been put forward as one of the public health priorities in Latvia, including emphasising that human milk is one of the most important factors in promoting children’s health [26], breastfeeding rates in Latvia are critically low. Statistics show that, in 2022, only ~16% of infants in Latvia were exclusively breastfed for the first six months [27].

Overall, the European region has the lowest rates of exclusive breastfeeding at six months—only approximately 25% [28]. Although national governments should commit to evidence-based promotion activities, including financial and political support, to improve breastfeeding rates [28], the consumption of commercial infant formulae and follow-on formulae is increasing yearly, correlating with the duration of breastfeeding decrease [29]. The commercial milk formula industry influences not only healthcare professionals’ and policy makers’ decisions and attitudes towards breastfeeding, but the use of infant formulae and follow-on formulae without proper indications from healthcare professionals is also being more and more normalised in society [29].

Despite the fact that it is prohibited, the Consumer Rights Protection Centre in Latvia regularly observes that the commercial milk formula industry provides social media creators with gifts or free samples to promote specific brand infant formula or follow-on formula to their followers [30].

To promote breastfeeding, parents, healthcare professionals, and society should be informed not only of the benefits of breastfeeding but also of the risks of infant formulae and follow-on formulae use [31], e.g., the risks that infant formulae can contain a significantly higher content of some trace elements compared to human milk [13]. A study from Spain [13] reports that the aluminium concentration difference in starter infant formula compared to full-term mature human milk is 86%, the nickel concentration difference is 58%, and the tin concentration difference is 72%.

Although food frequency intake data was collected during the study, as the majority of values for heavy metals and trace elements in human milk were below the detection limit, we could not further evaluate if the consumption of specific food products and drinks impacts the heavy metal and trace element content in human milk among breastfeeding women in Latvia. Data from other studies indicate that more frequent fish consumption is associated with a higher arsenic concentration in human milk [3,12]. Nevertheless, fish consumption among the participants of this study was low—on average, only once a week, which is lower than recommended for women during the lactation period [32]. Researchers from Norway [33] have reported that lead content in human milk is associated with the intake of by-products from wild animals (liver and kidneys). Researchers from Slovenia [34] have reported an association between cadmium concentration in human milk and vegetable intake. Nevertheless, only a small proportion of participants in this study (*n* = 13) reported consuming vegetables daily, as advised by national recommendations for lactating women [32].

Only a few studies from Latvia have analysed the content of heavy and trace elements in various foodstuffs such as vegetables, berries, potatoes, honey, dairy products, and eggs [35,36,37,38,39]. Therefore, one can only speculate whether these food products could also be the dominant dietary sources of these elements’ presence in human milk among breastfeeding women in Latvia.

Other factors can also influence the concentration of heavy metals and trace elements in human milk. For example, a higher arsenic concentration in human milk is found among women residing in urban areas, and lead concentration in human milk is associated with smoking, frequent use of cosmetic powders and lipstick, and the use of well water [13,40]. Part of the population who lives in rural areas of Latvia still uses water from wells [41]. However, a study conducted in the 1990s [42] already indicated a minimal impact of industrial pollution on well water quality in Latvia, and the impact is probably even lower at present. Comprehensive well water testing revealed the excessive presence of lead, cadmium, and nickel only in less than 1% of the wells in Latvia [42].

However, it is necessary to continue monitoring the pollution levels in Latvia. In future studies, a more comprehensive participant survey should be conducted in order to better evaluate pollutant presence and their sources in the human milk of women living in Latvia.

## 5. Conclusions

Obtained preliminary data show that human milk among lactating women in Latvia contains only insignificant amounts of heavy metals and trace elements. This should not be a reason to choose formula feeding over breastfeeding. Nevertheless, the content of heavy metals, trace elements, and other pollutants in human milk should be continuously monitored.

## Figures and Tables

**Figure 1 nutrients-16-01568-f001:**
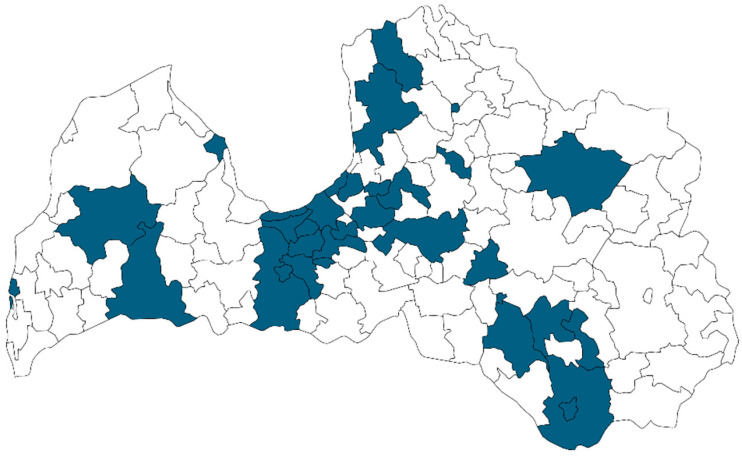
Map of Latvia. Regions where samples were collected are marked in blue (*n* = 116).

**Figure 2 nutrients-16-01568-f002:**
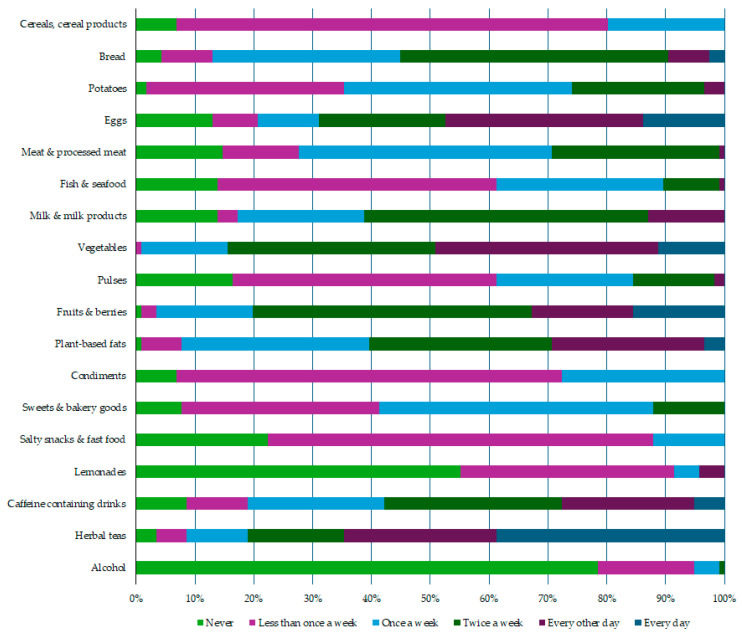
Evaluation of the dietary habits. Data from the food frequency questionnaires (*n* = 116).

**Table 1 nutrients-16-01568-t001:** Characteristics of the Participants.

	Year 2016–2017, *n* = 66	Year 2022–2023, *n* = 50
Characteristics	Mean ± Standard Deviation (Range)	
Maternal characteristics		
Age (years)	31 ± 4 (23–39)	30 ± 4 (21–40)
Maternal body mass index ^1^	22.33 ± 3.22 (17.85–32.18)	23.14 ± 7.00 (17.82–29.07)
Parity	2 ± 1 (1–4)	1 ± 1 (1–4)
Breastfeeding pattern	39—exclusive breastfeeding, 27—partial breastfeeding (two participants combining breastfeeding and formula feeding, 25 participants combining breastfeeding with complementary feeding)	50—exclusive breastfeeding
Milk expression manner	19—by hand 38—using a breast pump 9—combining both methods	11—by hand 31—using a breast pump 8—combining both methods
Child characteristics		
Age (months)	6 ± 4 (2–21)	3 ± 1 (1–5)
Sex	31—female, 35—males	23—females, 27—males
Birth weight (kg)	3.51 ± 0.59 (1.60–5.36)	3.50 ± 0.44 (2.70–4.60)
Birth length (cm)	53 ± 3 (42–61)	53 ± 3 (49–60)

^1^ Anthropometric measurements were not performed in this study. Body mass index was calculated based on given values by the participants.

**Table 2 nutrients-16-01568-t002:** The Content of Heavy Metals and Trace Elements in Obtained Human Milk Samples.

Element (Symbol)	Detection Limit (mg kg^−1^)	Year 2016–2017, *n* = 66	Year 2022–2023, *n* = 50
		Mean (mg kg^−1^) (Range)	
Arsenic (As)	0.005	<0.005 (<0.005–0.009)	not analysed
Cadmium (Cd)	0.005	<0.005	<0.005
Lead (Pb)	0.010	<0.010 (<0.010–0.047)	<0.010
Aluminium (Al)	5	<5	not analysed
Tin (Sn)	0.5	<0.5	not analysed
Nickel (Ni)	0.5	<0.5	not analysed

**Table 3 nutrients-16-01568-t003:** Differences in the Dietary Habits of Participants, Compared by Study Periods.

Food Products and Drinks Categories ^1^	Year 2016–2017, *n* = 66	Year 2022–2023, *n* = 50	*p*-Value
	Mean ± Standard Deviation (Range)		
Cereals, cereal products	1.02 ± 0.44 (0–2)	1.30 ± 0.43 (0–2)	0.031
Eggs	2.62 ± 1.54 (0–5)	3.42 ± 1.53 (0–5)	0.003
Sweets and bakery goods	1.40 ± 0.62 (0–3)	2.01 ± 0.72 (0–3)	0.002
Alcohol	0.48 ± 0.71 (0–3)	0.00 ± 0.00 (0–0)	0.000

^1^ Frequency of consumption was categorised as follows—never (0), less than once a week (1), once a week (2), two times a week (3), every other day (4), every day (5).

**Table 4 nutrients-16-01568-t004:** Comparison of Heavy Metal and Trace Element Concentration in Human Milk (μg L^−1^).

Element (Symbol)	This Study, Latvia	Björklund et al. (2012), Sweden [12]	Mandiá et al. (2021), Spain [13]
Arsenic (As)	<5 (<5–9.27)	0.55 ± 0.70 (0.04–4.60)	1.37 ± 1.82 (0.93–1.82)
Cadmium (Cd)	<5	0.09 ± 0.05 (0.03–0.27)	0.15 ± 0.20 (0.10–0.20)
Lead (Pb)	<10 (<10–48.41)	1.50 ± 0.90 (0.74–6.40)	0.30 ± 0.23 (0.25–0.36)
Aluminium (Al)	<5000	185.00 ± 584.00 (21.00–4393.00)	7.29 ± 1.11 (7.02–7.56)
Tin (Sn)	<500	0.40 ± 0.10 (0.21–0.77)	0.07 ± 0.00 (0.07–0.07)
Nickel (Ni)	<500	0.96 ± 6.5 (<0.085–47.00)	2.35 ± 2.69 (1.69–3.00)

**Table 5 nutrients-16-01568-t005:** Results of Heavy Metal and Trace Element Content in Human Milk from this Study in Comparison with Maximum Levels Defined in Legislation for the Production of Infant and Follow-on Formulae.

Element (Symbol)	Mean (mg kg^−1^) (Range)	Maximum Level Set in the Commission Regulation (CE) No. 2023/915 (mg kg^−1^) [25]
Arsenic (As)	<0.005 (<0.005–0.009)	0.020 (infant formulae and follow-on formulae marketed as powder) 0.010 (infant formulae and follow-on formulae marketed as liquid)
Cadmium (Cd)	<0.005	0.010 (infant formulae, follow-on formulae marketed as powder and manufactured from cow’s milk proteins or from cow’s milk protein hydrolysates) 0.005 (infant formulae, follow-on formulae marketed as liquid and manufactured from cow’s milk proteins or from cow’s milk protein hydrolysates)
Lead (Pb)	<0.010 (<0.010–0.047)	0.020 (infant formulae and follow-on formulae marketed as powder) 0.010 (infant formulae and follow-on formulae marketed as liquid)
Aluminium (Al)	<5	Not defined
Tin (Sn)	<0.5	50 (canned infant formulae and follow-on formulae, including infant milk and follow-on milk, excluding dried and powdered products)
Nickel (Ni)	<0.5	Not defined

## Data Availability

The data supporting this study’s findings are available on request from the corresponding author [L.A.]. The data are not publicly available because they contain information that could compromise the privacy of research participants.

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
