# Peer review of "Assessment of Heavy Metals and Trace Elements in the Human Milk of Women Living in Latvia and an Evaluation of Influencing Factors"

_nutrients, 2024, doi:10.3390/nu16111568_

Round 1

Reviewer 1 Report

Comments and Suggestions for Authors

Interesting paper, however I cannot see the reason for comparison of only 2 toxic elements  after 6 years' time, by the way they are called heavy metals.

Statistical analysis is very poor, only p value has been measured. The authors need to test their model, validate it and compare with other models. ANOVA would also help to test for significance.

We also have a food frequency questionnaire which should be seen in the appendix, which is not reflected in the title. This is very confusing.

Finally, English needs to be checked and discussion needs to be improved by comparison between similar studies.

Comments on the Quality of English Language

English needs to be checked

Author Response

We would like to thank the Reviewer for the detailed comments and suggestions provided for the improvement of the manuscript (nutrients-2994979). We believe that the comments have identified important areas which required improvement. After completion of the suggested edits, the revised manuscript has benefited from an improvement in the overall presentation and clarity. Below, you will find a point-by-point description of how each comment was addressed in the manuscript.  Original comments in regular typeface, responses in Italic.

Interesting paper, however I cannot see the reason for comparison of only 2 toxic elements  after 6 years' time, by the way they are called heavy metals.

Unfortunately, due to limited funding, we could not continue to determine the elements in human milk on the same scale.

Statistical analysis is very poor, only p value has been measured. The authors need to test their model, validate it and compare with other models. ANOVA would also help to test for significance.

As the majority of values for evaluated elements in human milk were below the detection limit, we were not able to provide further statistical analysis. To improve the quality of the manuscript, we have conducted additional statistical evaluation of obtained dietary data (L253-268).

We also have a food frequency questionnaire which should be seen in the appendix, which is not reflected in the title. This is very confusing.

We have corrected the title to the following – Assessment of Heavy Metals and Trace Elements in the Human Milk of Women Living in Latvia and Evaluation of Influencing Factors

Finally, English needs to be checked and discussion needs to be improved by comparison between similar studies.

Editing of the English was carried out. The Discussion section was improved

Reviewer 2 Report

Comments and Suggestions for Authors

The study presents important data in the area of ​​public health. However, the manuscript needs to be rewritten and reorganized. Some specific suggestions were made. There is a good methodology and adequate discussion of the results.

Abstract

L12-13: What is the difference between toxic and potentially toxic?

L12-14: Rewrite it to make it clearer, it's a bit repetitive and confusing.

L15-20: Rewrite it to make it clearer, it's a bit repetitive and confusing.

L22 keywords: use words that are not in the title.

Introduction: a presentation on contaminants, their risks, routes and concentrations were presented. But the important thing would be to bring studies that point out values ​​that are outside the acceptable range. Has this already happened? For what reason? What was the context? After this review, it would be necessary to highlight why this is being analyzed in Latvia. Present the novelty that justifies the publication of the data. It just looks like a report, with a diagnosis of the values ​​found.

Materials and Methods: You must evaluate whether the data follows the normal distribution and is parametric data. Only then can the appropriate statistical test be chosen.

Results

L202: Is the sample number (n), of 66 in 2016-2017 and 50 in 2022 and 2023, significant and representative for the country? Wouldn't it be appropriate to indicate the region or something like that? Is there this type of information? A map could help to understand better.

L217-240: These results did not appear in the abstract, discussion, conclusions and were not indicated as the objective of the study. I suggest adapting.

Author Response

We would like to thank the Reviewer for the detailed comments and suggestions provided for the improvement of the manuscript (nutrients-2994979). We believe that the comments have identified important areas which required improvement. After completion of the suggested edits, the revised manuscript has benefited from an improvement in the overall presentation and clarity. Below, you will find a point-by-point description of how each comment was addressed in the manuscript.  Original comments in regular typeface, responses in Italic.

Abstract

L12-13: What is the difference between toxic and potentially toxic?

As suggested by another Reviewer, definitions were changed to heavy metals and trace elements.

L12-14: Rewrite it to make it clearer, it's a bit repetitive and confusing.

The text was rewritten.

L15-20: Rewrite it to make it clearer, it's a bit repetitive and confusing.

The text was rewritten.

L22 keywords: use words that are not in the title.

Keywords were rewritten.

Introduction: a presentation on contaminants, their risks, routes and concentrations were presented. But the important thing would be to bring studies that point out values ​​that are outside the acceptable range. Has this already happened? For what reason? What was the context? After this review, it would be necessary to highlight why this is being analyzed in Latvia. Present the novelty that justifies the publication of the data. It just looks like a report, with a diagnosis of the values ​​found.

We have added some additional data, & information in the Introduction section about pollution sources in Latvia to justify the publication of the data.

Materials and Methods: You must evaluate whether the data follows the normal distribution and is parametric data. Only then can the appropriate statistical test be chosen.

We added clarification that due to small sample size, a non-parametric tests were used (L204-209).

Results

L202: Is the sample number (n), of 66 in 2016-2017 and 50 in 2022 and 2023, significant and representative for the country? Wouldn't it be appropriate to indicate the region or something like that? Is there this type of information? A map could help to understand better.

Population of Latvia was only 1875757 in 2022, with only 15954 live births (https://stat.gov.lv/lv/statistikas-temas/iedzivotaji/iedzivotaju-skaits/247-iedzivotaju-skaits-un-ta-izmainas). Breastfeeding rates currently also low in Latvia – only 9396 infants were breastfed till six months in 2022 (https://statistika.spkc.gov.lv/pxweb/lv/Health/Health__Mates_berna_veseliba/MCH100_kruts_barosana.px/table/tableViewLayout2/). For a small pilot study, 59 samples are defined as sufficient (https://www.crutzen.net/n.htm).

Latvia is a small country (64 589 km²), thus it would not be efficient to conduct research only in a certain region of Latvia. In this study, samples were collected from all over Latvia, without distinguishing a specific region for sampling.

L217-240: These results did not appear in the abstract, discussion, conclusions and were not indicated as the objective of the study. I suggest adapting.

The manuscript was replenished regarding the data of dietary habits. However, we would like to leave the main emphasis of the manuscript on the evaluation of heavy metals and trace elements in human milk.

Reviewer 3 Report

Comments and Suggestions for Authors

Breast milk is the most natural, safest and healthiest food, which is of great significance to promote the growth and development of infants and young children, but this does not mean that all breast milk is safe. The assessment of toxic and potentially toxic elements in Latvian women's breast milk is of great practical significance. However, the manuscript has the following problems:

1. The purpose of this paper is to assess toxic and potentially toxic elements in the breast milk of Latvian women. According to public information, in 2020, the birth rate of Latvia is 9.2‰, with about 17,000 births, and the number of samples selected in the manuscript was 66/50, which only accounts for about 0.35%/0.29% of the total number of pregnant women in that year. The number of samples selected is too small and unrepresentative. The sample size should be further increased.

 2. The breast milk sample in the lactation period needs to be clear, because the lactation stage will affect the result.

 3. The biggest problem of this paper is that the analysis process is too simple to draw a correct conclusion. As we all know, heavy metal detection technology is very mature, there is no updated method to supplement. This paper is aimed at the evaluation of toxic and potentially toxic elements in breast milk, and the emphasis should be on statistical analysis of evaluation methods. The factors affecting heavy metals in breast milk mainly include diet, living and working environment, cigarettes, cosmetics and so on. Statistical methods such as Logistic regression analysis and linear regression should be used to establish the relationship between heavy metal content in breast milk and diet, environment, cigarettes, cosmetics, etc.

Comments on the Quality of English Language

Minor editing of English language required.

Author Response

We would like to thank the Reviewer for the detailed comments and suggestions provided for the improvement of the manuscript (nutrients-2994979). We believe that the comments have identified important areas which required improvement. After completion of the suggested edits, the revised manuscript has benefited from an improvement in the overall presentation and clarity. Below, you will find a point-by-point description of how each comment was addressed in the manuscript.  Original comments in regular typeface, responses in Italic.

  1. The purpose of this paper is to assess toxic and potentially toxic elements in the breast milk of Latvian women. According to public information, in 2020, the birth rate of Latvia is 9.2‰, with about 17,000 births, and the number of samples selected in the manuscript was 66/50, which only accounts for about 0.35%/0.29% of the total number of pregnant women in that year. The number of samples selected is too small and unrepresentative. The sample size should be further increased.

The population of Latvia was only 1875757 in 2022, with only 15954 live births (https://stat.gov.lv/lv/statistikas-temas/iedzivotaji/iedzivotaju-skaits/247-iedzivotaju-skaits-un-ta-izmainas). Breastfeeding rates in Latvia is even lower – only 9396 infants were breastfed till six months in 2022 (https://statistika.spkc.gov.lv/pxweb/lv/Health/Health__Mates_berna_veseliba/MCH100_kruts_barosana.px/table/tableViewLayout2/). For a small pilot study, 59 samples are defined as sufficient (https://www.crutzen.net/n.htm).

Conducting research is only possible with financial support. To carry out a wider study and increase sample size, it is necessary to attract funding and modify the ICP-MS method to determine the content of elements in lower concentrations in biological samples.

  1. The breast milk sample in the lactation period needs to be clear, because the lactation stage will affect the result.

One of the inclusion criteria for participants was – at least 28 days postpartum. Therefore, only mature human milk samples were collected. Additional information was added to the "Materials and Methods" section to clarify this.

  1. The biggest problem of this paper is that the analysis process is too simple to draw a correct conclusion. As we all know, heavy metal detection technology is very mature, there is no updated method to supplement. This paper is aimed at the evaluation of toxic and potentially toxic elements in breast milk, and the emphasis should be on statistical analysis of evaluation methods. The factors affecting heavy metals in breast milk mainly include diet, living and working environment, cigarettes, cosmetics and so on. Statistical methods such as Logistic regression analysis and linear regression should be used to establish the relationship between heavy metal content in breast milk and diet, environment, cigarettes, cosmetics, etc.

As the majority of values for evaluated elements in human milk were below the detection limit, we were not able to provide further statistical analysis. To improve the quality of the manuscript, we have conducted additional statistical evaluation of obtained dietary data (L253-268).

Minor editing of English language required.

Editing of the English was carried out.

Round 2

Reviewer 1 Report

Comments and Suggestions for Authors

authors have replied sufficiently and paper can be accepted

Author Response

We thank the Reviewer for the second review of the manuscript (nutrients-2994979) and the positive assessment.

Reviewer 2 Report

Comments and Suggestions for Authors

All suggestions and recommendations were accepted. Only one issue needs to be adjusted:

L119-120. "Samples were collected from all over Latvia without distinguishing a specific region." -> Insert a map (location).

Author Response

We thank the Reviewer for the second review of the manuscript (nutrients-2994979). Below, you will find how the last issue was addressed in the manuscript.  Original comment in regular typeface, responses in Bold and Italic.

All suggestions and recommendations were accepted. Only one issue needs to be adjusted:

L119-120. "Samples were collected from all over Latvia without distinguishing a specific region." -> Insert a map (location).

We have inserted a map in the manuscript, providing information about the regions of Latvia where human milk samples were collected (Figure 1).

Reviewer 3 Report

Comments and Suggestions for Authors

Breast milk is of great significance in promoting the growth and development of infants. The assessment of toxic and potentially toxic elements in Latvian women's breast milk is of great practical significance. A new material needs to be added to the paper.

Due to the small sample size, the test results are not accurate enough to be representative when SPSS is used, and it is easy to lead to sampling bias. In this paper, Kruskal-Wallis H test is used to avoid the accuracy problems caused by small sample size, which is desirable. It should also be necessary to add sampling results of samples in the manuscript to check sampling deviations. According to the correlation between diet, life and other key factors and breast milk quality established in the manuscript, the breast milk quality of sampled samples would be predicted and compared with the results of ICP-MS to determine sampling bias.

Comments on the Quality of English Language

Minor editing of English language required.

Author Response

We thank the Reviewer for the second review of the manuscript (nutrients-2994979). Below, you will find how the comments were addressed in the manuscript. Original comment in regular typeface, responses in Bold and Italic.

Breast milk is of great significance in promoting the growth and development of infants. The assessment of toxic and potentially toxic elements in Latvian women's breast milk is of great practical significance. A new material needs to be added to the paper.

Due to the small sample size, the test results are not accurate enough to be representative when SPSS is used, and it is easy to lead to sampling bias. In this paper, Kruskal-Wallis H test is used to avoid the accuracy problems caused by small sample size, which is desirable. It should also be necessary to add sampling results of samples in the manuscript to check sampling deviations. According to the correlation between diet, life and other key factors and breast milk quality established in the manuscript, the breast milk quality of sampled samples would be predicted and compared with the results of ICP-MS to determine sampling bias.

We have added information to the manuscript regarding the drop-out rates and reasons for dropping out of the study. Only 5 participants did not complete the study due to the inability to collect the required amount of human milk for the analysis, and only 4 participants did not complete the study due to illness or illness of the child. We did not collect the questionnaires from those participants (as questionnaires were collected at the same time as collecting human milk samples from the participants). Therefore, we do not have information to evaluate sampling bias comparing those participants who did not complete the study and participants who completed the study.

Minor editing of English language required.

Minor editing of English language was provided.